# Resilience Through Belonging: Schools’ Role in Promoting the Mental Health and Well-Being of Children and Young People

**DOI:** 10.3390/bs15101421

**Published:** 2025-10-19

**Authors:** Ceri Brown, Alison Douthwaite, Michael Donnelly, Yusuf Damilola Olaniyan

**Affiliations:** Department of Education, Faculty of Humanities and Social Sciences, University of Bath, Bath BA2 7AY, UK; c.l.brown@bath.ac.uk (C.B.); ydo22@bath.ac.uk (Y.D.O.)

**Keywords:** belonging, well-being, young people, schools, identity

## Abstract

After almost a decade of the policy promotion of resilience-building initiatives in schools, mental health figures have not improved. Recent research points to shortfalls in adopting individualistic policy perspectives on resilience when supporting children’s well-being. The originality of this paper lies in our argument that a broader consideration of resilience that acknowledges the importance of collective and relational approaches could enhance school-based well-being support. Our social identities are multiple, and the individual aspects of our identities are multilayered. A more nuanced consideration of children’s sense of belonging across the different social domains of their lives is, therefore, important in developing well-being approaches that prevent poor mental health outcomes for all children. We draw on data from the Belonging in School Study, the largest school-belonging study undertaken in the city of London, which was conducted between 2023 and 2024. This paper focuses on data collected from online survey responses from almost 2000 students and subsequent focus groups with 76 students. Three identity factors emerge as highly important to young people’s sense of belonging in school: social and cultural identity, peer group identity and individual identity. Descriptive statistical analysis of the survey findings and thematic analysis of the focus group discussion suggest that the importance of the elements within these three factors is relative to students’ social characteristics and levels of social privilege.

## 1. Introduction

Schools’ renewed role in promoting student well-being has become a defining feature of 21st century education policy-making, especially since the pandemic. Rising mental health figures in the early 2010s have led to a fundamental shift away from a ‘pathogenic’ approach, which views health as ‘the absence of disease or illness’ ([69]), and towards a ‘salutogenic’ approach, which focuses on the whole student population (not just those at-risk) through health-promoting factors, aiming to facilitate the well-being of the whole child ([6]). This has placed schools’ role in mental health prevention front and centre in national governments’ ([131]) and the European Union’s strategies ([26]). Moreover, a focus on children’s well-being is increasingly viewed as a crucial element in addressing other public policy challenges, including educational attainment, youth unemployment and social exclusion. This ambition has been amplified since the COVID-19 pandemic, whose key legacy burden was on children’s mental health and well-being ([8]; [126]), with levels of anxiety and depression having doubled in a number of European countries ([88]).

Further international stressors including multiple global conflicts, mass displacement, environment crises and trading wars causing unstable major economies have resulted in a very real sense of insecurity, uncertainty and distrust for children and young people in even the wealthiest nations. Indeed, it could be argued that children born in the 21st century are growing up in one of the more precarious social epochs of modern history. Terms such as ‘I[phone] Gen[eration]’ ([122]) and ‘snowflake generation’ ([49]) have been used to describe today’s younger population, who is seen to be emotionally unable to cope with the ‘everyday realities of life’ in becoming more insular and individuated.

Within the UK, and echoing other national responses, the English government has aligned with such cultural commentaries, arguing that today’s young people are not psychologically prepared for the threats that they face in daily life. Education Secretary Bridgette Phillipson and Health Minister Wes Streeting have recently proposed that schools urgently need to employ measures to “cultivate much-needed grit amongst the next generation” ([94]). The assumption is that individual inabilities to cope are also the cause of other educational challenges, for example, “the triple threat of attendance, behaviour and mental health” ([94]). This deficit understanding of young people positions children’s own aptitudes for coping with societal stressors as the object of focus, continuing a cross-party policy trend on ‘toughening up’ young people throughout the 21st century. Nicky Morgan, former Education Secretary under the Conservative Government, was an early advocate of this ambition, defining grit as ‘the confidence and determination not to be beaten. It’s that attitude that says, “dust yourself off and try again”’ ([83]). To this end, she championed a Character Education approach, with the aim that schools should “instil characteristics in their pupils such as drive, grit and optimism’ ([133]). In earlier work (see [42]), we found that England, compared to other nations of the UK, had the most individualising and moralising approaches to social and emotional learning in schools, for example, in taking a ‘skills and competencies’ as well as a ‘moral values’ lens on well-being promotion, such as through the Character and Resilience Manifesto ([91]).

Resilience is a concept closely aligned with grit and determination that has been central to early policy discourses of how schools should promote children’s mental health in the UK (see [38]; [36]; [42]). Resilience, now identified as a key component informing the recent ‘personal development’ judgement by the English national school inspection framework ([89]), has been defined very narrowly in policy in terms of the individual skills and aptitudes needed to deal with societal challenges: “Firstly, a sense of self-esteem and confidence; secondly a belief in one’s own self-efficacy and ability to deal with change and adaptation; and thirdly, a repertoire of social problem-solving approaches” ([38]).

### 1.1. Widening the Resilience Lens in Educational Policy

After almost a decade of the policy promotion of resilience-building initiatives in schools, mental health figures have not improved. Indeed, some measures indicate that they have worsened ([88]). Importantly, a body of research in the UK has explored the effectiveness of school well-being policies founded on individual resilience ([18]; [19]; [20]; [23]). A review of educational policy approaches to social and emotional well-being in the UK identified a disproportionate dominance of individualistic approaches across all four nations of the UK based on strengthening children’s skills and competencies ([20]). The individualistic approach has also been seen to dominate among the mental health approaches that schools are advised to take in other Western contexts such as Australia and the United States ([23]).

A number of shortfalls in adopting an individualistic policy perspective to support children’s mental health have been identified. Schoolchildren have found the resilience narrative unhelpful and, indeed, harmful to their mental health ([19]). Through methods tasking young people across multiple school sites to photograph and then discuss the ways mental health was conceptualised in schools, this study identified a common narrative aligning schooling narratives about resilience with policy discourses on individual psychological strength and coping. Indeed, schools were found to conflate health with educational achievement and place the responsibility for students’ mental health squarely onto their own shoulders, which prevented them from reaching out and seeking support. Young people themselves, however, have called for school well-being approaches that recognise the importance of their relationships in mental health and value wider aspects of their skills and aptitudes so that students feel they can be themselves ([23]).

This points to a need to reconfigure school well-being approaches away from measures seeking to individualise responses and towards broader definitions of resilience that seek collaborative and social approaches. Three distinct theoretical positions are notable among school-based mental health approaches that have sought to strengthen children’s mental health and well-being (see [65], for a review). The first and most prominent of these is PERMA. Developed from the positive psychology tradition pioneered by [111] ([111], [112]), this approach has been advanced within a number of high-profile school-based study interventions. For example, the *Geelong Grammar School* programme in Australia embedded the PERMA model across curriculum, pedagogy and whole-school practices to promote student resilience and well-being ([87]; [86]). Similarly, [111] ([111], [112]) describes large-scale trials of the *Penn Resiliency Program (PRP)*, which adapted positive psychology techniques into structured school lessons aimed at improving students’ optimism, emotional regulation and problem-solving skills. From a PERMA perspective, relationships are considered important for the child from an individualistic perspective—how the relationships benefit the child’s well-being and belonging. The second approach, Self-Determination Theory, developed by [110] ([110]), is a theory of motivation that explains children’s well-being in schools. It has also underpinned a number of school interventions ([105]; [118]). From this perspective, relationships are still conceived of as an individualistic gain, but in a less performative sense and with the recognition that the gains are also enjoyed through the friendship group. In contrast to the former psychological approaches, the third approach is sociological and termed Relational Well-being ([134]), which has enjoyed recent development into schooling interventions ([22]). From this perspective, well-being is conceptualised as a shared collective and contextual endeavour which, we argue, offers greater potential to address the individualistic shortcomings of current policy approaches by reframing a well-being focus away from individual resilience and coping skills and towards a focus on relationships and belonging, which is in line with the social and community framing of resilience advanced by scholars seeking to address the limitations of individualistic resilience narratives (see [31]; [30]).

More recently, there have been signs that education policy-makers outside of England are starting to recognise the importance of collective and relational approaches. Wales, for example, recognised the importance of an identity-based approach in its aim to strengthen Welsh identity culture and language (Curriculum for Wales, 2022, Education for Sustainable Development and Global Citizenship), while Northern Ireland showed elements of an ethical values approach aiming to strengthen community cohesion as central to community well-being (Children and Young People Strategy, 2017–2027). Scotland, by contrast, recognised the socio-economic gradient to well-being (Curriculum for Excellence, Getting it Right for Every Child) (see [41]).

At the regional level within England, local governments have sought to strengthen community resilience within place-based policy initiatives aiming to stimulate educational outcomes, employment and life chances. For instance, in Manchester’s Building Stronger Communities Together Strategy 2023–2026, Councillor Joanna Midgley, Manchester City Council’s Deputy Leader, explains the importance of relationships and belonging on life chances:
“We have launched this strategy with the aim of forging even stronger bonds across our communities, and creating a stable foundation for future prosperity”.([11])

Another example is the Mayor of London’s Violence Reduction Unit’s (VRU) Belonging in School Study ([21]), with a primary aim of capturing students’ perspectives on what they need from education to feel the sense of belonging and safety necessary to be included and succeed at school. This research informed key principles for inclusive education—‘Embedding Equity and Diversity’, ‘Students as Active Citizens’, ‘Being Adaptable and Reflective’ and ‘Beyond Academic Achievement’—that led to the VRU’s development of London’s Inclusion Charter ([129]). This paper reports on the findings from this major large-scale study, drawing on evidence from over 2000 school students from 81 schools across each of the 32 boroughs of Greater London.

In what follows, we review the evidence for how belonging in schools may be a more effective approach to promoting children’s mental health and well-being than the current individual resilience policy approach. We then explain the methodology by which the data on the key constituents of belonging for students was derived, drawing on both survey data and semi-structured focus groups. In the following section, we present evidence to demonstrate the centrality of identity-related elements to belonging and end with a discussion on what this could mean for the future direction of schools’ role in supporting child well-being.

### 1.2. Belonging, Well-Being and Identity

We define belonging as an affective and cognitive state that encompasses the interplay between the social and the individual aspects of identity.
“That sense of being somewhere where you can be confident that you will fit in and feel safe in your identity, a feeling of being at home in a place, and of being valued.”([103])

Following [134] ([134]), we recognise belonging as being fundamentally contextual in being wholly relative to a particular place, space and set of relations—it is, in essence, relational. This may be why educationalists have primarily focused on school belonging as the primary mechanism for children’s well-being ([60]; [70]), involving identification with school and a sense of membership with the school community ([2]).

However, given that our social identities are multiple and that the individual aspects of our identities are multilayered, recent research has argued that in order to understand the complex process of identity construction that is negotiated interpersonally, socially and spatially, schools’ role should go beyond the school environment, and the role of other social domains should be considered such as the family, peers and community ([78]; [92]; [108]).

At the heart of it, there are three central propositions that unify recent research on schools’ role in supporting children’s belonging. Firstly, well-being approaches must be fundamentally relational in their approach, founded on principles of care, trust and identity (see [85]). Secondly, belonging must extend beyond the school gates and involve attending to the various domains of children’s daily lives—school, social life, local community and wider society ([19]; [106]; [135]; [31]; [30])—to enhance resilience and support student well-being ([67]). Thirdly, the school’s role in building and affirming the social and individual aspects of students’ identities is central to their well-being ([7]; [10]; [128]).

Social identities refer to the extent to which we identify and feel a sense of membership with others (that is, the ways in which we feel similar, share values and feel connected with others), while individual aspects of our identity refer to the need to feel unique, special and distinct from others. The work of [44] ([44]) is important in delineating further identity, in terms of both individual skills and characteristics as well as those related to perceptions about values and expectations. [44] ([44]) contends that, taken together, these aspects of identity are the key drivers of motivated action. As well as supporting the case for the importance of considering both personal and collective identities when conceptualising school well-being support, this also has implications for better understanding why different demographic groups experience differing levels of belonging and success in schools. While belonging may primarily be a subjective and perceptual state in that the individual must ‘feel’ that they belong, this is contingent upon being recognised as belonging by others ([56]). A comprehensive sense of belonging, therefore, must simultaneously strengthen both our social and individual identity needs.

The findings from the Mayor of London’s Belonging in School Study (see [21]) highlighted the centrality of identity for students’ sense of belonging and safety in school. In this paper, we explore the various constituents of identity that different groups of students identified as being the most significant for their sense of belonging and canvass a number of reasons as to why this may be the case.

## 2. Methodology

The Belonging in School Study was a major piece of mixed-methods research carried out in the city of London between 2024 and 2024 using a Sequential Exploratory Design ([66]). In line with [58]’ ([58]) realist research paradigm, the initial quantitative element of the research was aimed at identifying observable phenomena. The following qualitative phase then aimed to deepen understandings of the causal mechanisms underpinning the phenomena. In the case of our research, for example, observable phenomena might be particular social groups being more likely to exhibit particular behaviours or perceptions, with the qualitative data deepening understandings about why that might be the case. The presentation and analyses of results follow this approach, with quantitative data used to signal phenomena for further qualitative analyses, as opposed to an integrative approach.

### 2.1. Setting

This research aimed to identify the factors that school students in London believed to be the most important for their well-being and consider their reasons underpinning these assumptions. In total, 81 schools representing at least 1 from each of the 32 boroughs of Greater London took part. Local Authority and Inclusion partner networks were mobilised to identify a broad range of school and educational providers (including alternative provisions and Pupil Referral Units (PRUs)). Participation was voluntary and anonymous, with parents and students providing informed consent. This study was approved by the University Ethics Committee. We intentionally recruited schools with diverse student populations in terms of socio-economic background, SEND and ethnicity to ensure representation.

### 2.2. Measures

The first phase of this study invited secondary-school-aged children to complete an anonymous online survey administered via school networks. This asked them to rate the importance of 17 belonging-related factors. The rating scale for each item was 0 (not important), 1 (slightly important), 2 (quite important), 3 (important), 4 (very important) and 5 (extremely important). The list of factors was informed by an in-depth review of the literature on school belonging to ensure that the survey factors were comprehensive and empirically validated. Each survey item was displayed on its own screen, with an image chosen to reflect the concept in question. These design features helped make the survey more widely accessible by reinforcing the verbal description with a visual element and by reducing the amount of information students had to process within each screen. A full list of survey items, with their corresponding literature sources, is provided in Table 1.

### 2.3. Participants

A total of 2078 secondary-aged students (aged approximately 11–18) across Greater London completed the online survey. Of these, 76 students took part in follow-up focus groups. The survey collected information on key social characteristics that may shape experiences of belonging, including gender, sexuality, faith, disability and/or special educational needs (SEND), neurodivergence, care experience, young-carer responsibilities, refugee or asylum-seeker status and English as an Additional Language. In addition, anonymised administrative data to indicate socio-economic status (Free School Meal eligibility and Pupil Premium status) were linked to student responses via a unique identifier. This enabled us to examine belonging across diverse groups, including students from non-dominant and marginalised backgrounds. Table 2 provides an overview of the covariates collected and linked to the survey.

### 2.4. Procedures

Qualitative data was generated in the second phase of the study through focus groups. We prioritised schools and alternative providers in boroughs with high levels of deprivation and diversity. To enhance accessibility and participation, focus groups were conducted in a range of formats (single school, cross-borough, in person and online). Focus groups were typically 45–60 min, facilitated by trained researchers, audio-recorded with consent and professionally transcribed.

The survey and focus group questions were co-developed with the VRU’s Youth Participation Advisory Group (YPAG) to ensure inclusivity, accessibility and contextual appropriateness. While direct member-checking with participants was not feasible, YPAG acted as a youth validity check, which enhanced credibility and supported the contextual relevance of interpretations.

### 2.5. Data Analysis

Applying a sequential explanatory design ([66]), we first analysed quantitative survey data to identify key belonging factors and subgroup differences. These results then guided the design of the qualitative focus groups so that students’ explanations and lived experiences contextualised the survey patterns. Integration of the two phases occurred during analysis and interpretation.

Quantitative survey data were analysed using descriptive statistics in order to identify the relative importance of the 17 belonging-related factors across the whole cohort. Mean scores were calculated for each factor, and rank orders were established to determine their comparative significance. Analyses were disaggregated by key demographic variables (e.g., gender, socio-economic status, SEND, ethnicity, EAL, care experience, refugee/asylum status, sexuality and faith) to explore variation across social groups. To establish the “top ten” factors, we ordered the mean score position for each social group. The mean position is the ordering of mean scores from smallest to largest in order to identify patterns in the data. This allows us to identify both the overall priorities and those specific to non-dominant groups. All analyses were descriptive rather than inferential, reflecting the exploratory and inclusive nature of the study design. Analyses were carried out in Microsoft Excel and SPSS (v.29).

The qualitative phase involved thematic analysis ([51]; [47]) of the focus group discussions. Two members of the research team independently coded the data before cross-checking and refining themes to enhance reliability. Coding was both deductive (informed by the survey factors and relevant literature on belonging) and inductive (allowing new themes to emerge from students’ accounts). To enhance reflexivity and transparency, the research team also kept an analytic journal, recording reflections and observations immediately after each focus group. These notes were revisited during coding to ensure that contextual insights informed theme development. Analytical decisions and theme refinement were then discussed with the wider research team and the VRU’s Youth Participation Advisory Group (YPAG) to ensure validity and contextual appropriateness. Coding and theme development were conducted manually, with transparency and reliability strengthened through independent double-coding, cross-checking and iterative team discussions.

## 3. Results

### 3.1. Data on the Important Factors That Support Students’ Sense of Belonging

Identity was clearly the most central factor leading to students’ secure sense of belonging in school, according to the 2078 participants in our online survey. Nine of the top ten factors (see Table 3) rated as the most important to students’ sense of belonging concerned issues of identity (see Table 4).

This led us to distinguish between the different types of identity (see Table 5) in considering their relative importance for students across the cohort and according to the different profiles of the students. While there are different aspects of identity and domains in which they are important (see [22]), we focus here on the three aspects of identity that were rated the highest and those most discussed within the focus groups: social and cultural identity, peer identity and individual identity (see Table 5).

### 3.2. Social and Cultural Identities

#### 3.2.1. Quantitative Evidence from the Online Survey About the Significance of Social and Cultural Identity for Belonging

Overall, factors that were related to students’ social and cultural identities were rated as the most important constituents of students’ sense of belonging in school across the whole secondary-school-aged cohort. Specifically, this referred to three of the top ten items rated within the survey. The first, ‘being treated with as much respect as everyone else in the school’, was the most important constituent of social and cultural identity and the overall highest rated factor, with a mean of 4.26 (where 4 is very important and 5 is extremely important). The third highest rated factor, ‘people from different backgrounds feeling welcome and heard in school’, was universally important for the students, with a mean of 4.01 suggesting the high importance afforded to social and cultural identity by children in London as central to supporting their sense of school belonging. Finally, the indicator ‘seeing different backgrounds represented in school’ was rated the sixth most important factor for belonging, with a mean of 3.76 (where 3 is important and 4 is very important).

Notwithstanding the universal importance of social and cultural identity for the students, it was notable that, for some groups, it was particularly important (see Table 6). This included socio-economically (SES) disadvantaged students, girls, those with English as an Additional Language (EAL), Black/Black British and Asian/Asian British students, who rated and/or ranked all three factors more highly than the overall. Here, it is notable that each of these characteristics indicates that students are from non-dominant social or cultural identity groups.

Interestingly, while ‘being treated with as much respect as everyone else in school’ was rated as more important (relative to other students) for Black/Black British, Muslim, Sikh and Christian children, for students who were Hindu, Atheists and those with ‘no religion’, this factor was rated less than the mean, while those same groups rated ‘people from different backgrounds feeling more welcome and heard’ more highly (relative to other groups). This suggests that for children without religious beliefs, or for those from minority religious communities in London, it is possible that they do not feel that their views are so welcome within their schools. Further research is needed on the impact of religious beliefs on children’s sense of belonging in school.

It is also interesting to consider those groups for whom social and cultural aspects of identity were consistently less important relative to other students. This included, for boys, students who were not socio-economically disadvantaged, white students and students from gender and sexual minorities. The relative importance of social and cultural identities was also lesser for some students who have SEND, are care-experienced or are refugees or asylum seekers.

#### 3.2.2. Qualitative Data on the Significance of Social and Cultural Identity for Belonging

The qualitative data highlighted three aspects that were important for students to generate a secure sense of social and cultural identity: respect, cultural awareness and visibility.

##### Respect

The aspect of respect in school was connected to themes of conflict, safety and aggression and was raised in focus groups mostly by girls. Respect was understood to be a necessary foundation for order and peace in school and was generally discussed in terms of instances where it was lacking. There was a broad consensus across participants that a respect for difference or diversity in thought or backgrounds was more important for maintaining a harmonious school environment than being liked or agreed with:

“You don’t have to like someone in order to respect them, and as long as everyone has respect for everyone, then that positive relationship is there regardless of whether or not you disagree with each other’s opinions.”FT, Student, Alternative Provision A

“So long as you can get through the day with respecting each other and without getting into fights, again you don’t need to have any interactions with them, completely avoidable.”MT, Student, Alternative Provision A

Students often associated being disrespected with intimidation or physical aggression, whereas some groups (girls in particular) described boys’ actions as dominating or encroaching on their sense of safety and belonging. One girl explained, “we had an issue previously when boys would just go around slapping each other’s heads…crossing a boundary… and then they end up doing it to other people.” Another girl described how “you’ll see aggression, even just like pushing in the corridors because it gets really packed sometimes so you just feel that sometimes you’re about to get knocked over.”

##### Cultural Awareness

Secondly, the qualitative data identified the importance of schools facilitating a meaningful understanding of power inequalities and a deep awareness of other cultural values, beliefs and traditions. This aspect connected with negative themes such as ignorance, prejudice, racism and fear of the unknown, as well as more positive ones including social justice, diversity and broad-/global-mindedness. It was particularly raised by students from ethnic minority groups, those with English as an Additional Language and those from refugee and migrant backgrounds. Students recognised that the curriculum was a powerful mechanism for either generating cultural awareness or obstructing it:

“I feel like there needs to be more like lessons on actually like different cultures and stuff because… I came to the school (from) a catholic school, so there’s only really like one religion and stuff, so I came here not really knowing much…I feel there needs to be more like diversity in the classes and stuff, like even about races, cultures, religions… because a lot of people like, they fear the unknown.”BL, Student, Secondary Mainstream A

A number of students from minority ethnic communities identified that the schools’ commitment to supporting social and cultural identity must be an ongoing and genuine endeavour, otherwise it can feel tokenistic. Where poorly taught, it could be perceived as legitimising the dominance of privileged groups or as a superficial approach to respecting diversity:

“When it comes to black history month… you shouldn’t only focus on it in one month (per year) … if they do it like yearly, consistently, I think that would… utilise diversity and like culture throughout.”CL, Student, Secondary Mainstream A

“there’s quite a toxic culture of like young people being like homophobic or racist or like sexist just because they’re not being taught about it properly… usually most schools just put on like another assembly, that half the kids go off in, so there needs to be a better way of like actually educating kids.”AW, Student, Secondary Mainstream B

Students gave multiple examples of ignorant, misogynistic or racist narratives in school to evidence the need for schools to facilitate a deeper understanding and the celebration of diversity, for example, “I know loads of cases with young people being like …racist or like sexist,” and mentioned “hearing a lot of like… like Andrew Tate and stuff,” particularly in the “younger years”.

##### Visibility

Finally, students mentioned the importance of non-dominant groups ‘taking up space’ in school, referring to the need to be visible and valued in the school environment. This aspect connected to themes such as power, free expression and the right to exist, which were dominant themes among students from low-income backgrounds. Students recognised that uniform policies can be restrictive and antithetical to the ability to ‘express yourself… especially like culturally.’ Visibility was also aided through exposure, with one girl observing how a critical mass within the school community normalised and gave her the freedom to embody cultural markers of dress and physical expression:

“Having a more diverse group of students is really nice because, for example, me, when I was younger, everyone around me they used to straighten their hair and then, like now, I’m friends with people who are more, you know, expressive with their culture and their hair so, you know, it just makes me feel more comfortable to do so.”AL Student, Secondary, Mainstream A

The focus group data suggested that some students from underprivileged backgrounds felt that their social or cultural identities were not recognised or valued by other teachers and students and that this obstructed their sense of belonging. These students argued that a generalised culture of respect and understanding was more significant for their sense of belonging than a sense of being universally liked or popular. One boy on free school meals observed that

“getting along with other students at school… getting along with someone isn’t as important as just having them respect you. Like you don’t have to converse with everyone that you meet. …, you don’t have to be best friends with everyone, you don’t have to talk to everyone. But just in the sense that like if you guys have a disagreement, you can talk it out respectfully and you guys have a respectful conversation. But getting along with other students to me isn’t as important. I just want to feel I’ve got the same rights to be here.”FT, Student, Secondary, Alternative Provision A

While social and cultural identity was universally recognised in the survey responses as important for school students, the qualitative data shed light on why it was deemed to be of relatively greater importance for girls, ethnic minorities and students from socio-economically disadvantaged backgrounds. This may be due to a sense that their cultural backgrounds are not understood or recognised or to a feeling of being oppressed by dominant and privileged groups in school and feeling less able to express their cultural identities with freedom and confidence.

### 3.3. Peer Group Identity

#### 3.3.1. Quantitative Evidence from the Online Survey About the Importance of Peer Group Identity to Belonging

Indicators that referred to students’ peer group identities were rated as the second most important constituent of belonging for the overall cohort of secondary-school-aged students in London.

Three factors spoke to students’ peer identities. The first was ‘having a friend or group of friends in school that I trust’, which was rated the second highest factor, with an overall cohort mean of 4.25. The 0.01% difference between this factor and the highest rated factor (being treated with as much respect as everyone else in school) indicates how closely aligned the two indicators were as constituents of students’ sense of belonging in school. Secondly, ‘Getting on with other children in school’ was rated as the fifth highest factor, with a mean rating of 3.82. Thirdly, ‘having a space where me and my friend can hang out in school’ was rated as the eighth most important aspect of belonging, with a mean of 3.64 across the whole cohort. As with social and cultural identity, the finding that these three measures were collectively rated within the top ten factors important for belonging indicates the generalised importance of children’s peer relationships and friendships in school.

As with social and cultural identities, the findings show that peer aspects of identity are universally important for schoolchildren. However, scrutiny of the relative importance of peer-related factors for different groups highlights some subtle but notable differences between those groups that ranked and rated peer identity factors higher than the norm and those who ranked and rated social and cultural identity factors lower than the overall cohort.

For the following groups, ‘having a friend or group of friends in school I trust’ was ranked as the single most important factor contributing to students’ sense of belonging in school: boys, students who were not socio-economically disadvantaged and white students. The same groups ranked or rated both other measures of peer identity more highly relative to other groups across the cohort. Here, it is notable that each of these characteristics reflects a more privileged social or cultural identity position.

However, other groups also ranked or rated two of the three peer identity factors highly, including young carers and, with slightly lower mean ratings, students with special educational needs and disabilities and asylum seekers.

However, there were some students who rated the ‘getting on well with other students in school’ factor particularly highly. For students with SEND, Transgender students and non-binary students, this was the highest rated of all the factors (in comparison with an overall ranking of fifth most important across the cohort). This suggests that a sense of belonging and identification with all peers is more important than their friendship relationships for these students. This finding is consistent with those of [46] ([46]), who identified that the *‘perceived similarity’* obtained by a sense of fitting in with or sharing experiences with peers in their educational setting was a key aspect of belonging, particularly where differences could be perceived negatively by peers.

It is also interesting to consider those groups for whom peer identity aspects were less important relative to the whole cohort. This included Black/Black British and Muslim students, who rated or ranked peer identity factors less important than the overall cohort, as well as Asian/Asian British students, who ranked or rated two of the three indicators lower than the rest of the cohort. This aligns with the findings on students’ social and cultural identities, which indicated that this aspect of their identity was more important than the peer identity aspects for students’ sense of belonging in school (Table 7).

#### 3.3.2. Qualitative Data on the Significance of Peer Identity for Belonging

The focus group data revealed a consensus among the students that seeing their friends was the most important aspect of coming to school. This was expressed both among those who enjoyed as well as those who did not enjoy school:

“I don’t necessarily like coming to school but I think my friends definitely make it better.” JC, Student, Secondary Mainstream C

“Well because let’s say you had no friends at school, you’re not going to want to come to school because you’ve got no one to speak to you. So, friends make school.” JD, Student, Secondary Mainstream D

“School revolves around like friend groups like friendships. Most teachers see it as being grades and like teaching, but most people see it as like another social life.” JC, Student, Secondary Mainstream E

In reflecting upon why boys’ rating for peer identity identified it as the most important aspect of belonging for students, we found it notable that, similarly to girls’ accounts, boys also observed a male dominance in school, and they identified friendships as a means by which to establish a reputation and status:

“we seem to have a very big problem with people, a certain dominance over new kids, because they feel like this is their environment and they shouldn’t let other people come in…especially the boys when they come in, they get pressured…like they look at you, assume that you are a threat to their reputation in school.” GT, Student, Alternative Provision A

This echoes girls’ accounts from a different focus group discussion on male dominance in school, this time identifying friendships and peer relationships as a means by which to establish a reputation and status. This echoes [125]’s ([125]) findings that dominant and bullying behaviours, particularly among adolescent boys, may be ‘fueled by the desperate need to belong.’ This highlights the need for a more nuanced understanding of the factors that support a sense of school belonging for different groups and consideration of how different forms of social and peer exclusion may play a part in bolstering a sense of belonging for some students. It suggests an interplay between social and cultural identities and peer identities in enabling or hampering a sense of belonging for students in school.

Students with SEND also ranked peer identity as the most significant aspect of belonging. The qualitative data suggested that for these students with non-cultural disadvantages, peer identity may be relatively more important than social and cultural aspects of their identity for a sense of belonging in school. Supporting other research ([100]; [73]; [13]), the findings suggested that this could be because of the protective status that friends and peers can bring for vulnerable children:

“I feel like a lot of jokes are made around especially SEN and then people… feel like they should like, ha ha, laugh at it as well.” CL, Student, Secondary Mainstream A

“if you meet someone nice in school they help you fit in in school and like introduce you to their friends as well which creates like, it’s kind of like a protective barrier from all the bad things in school and like they help you in general with your social skills and like becoming friends with other people.” AW Student, Secondary Mainstream B

Notwithstanding the importance of friendship to students’ sense of belonging and general schooling experience, the students observed that not all friendships were equal. It was the quality and authenticity of friendships that were revered because of the affordance they offered for genuine expression:

“I feel like you should just be around people who don’t peer push you…Yes, so you should really build quality friendships.” HC, Student, Secondary Mainstream C

“I feel like all of this chatting about friendship it’s not really that important because let’s be real yeah, let’s be real, like often your friendships at school are fake and like not really your true self because you’re scared to show it.” BT, Student, Secondary Mainstream D

These comments highlight another interesting interplay between peer identity and individual identity in developing a sense of belonging and well-being at school. Genuine friendships and being able to feel safe to be yourself amongst peers are contrasted with aspects of peer identity that are harmful for belonging, namely, adopting friendships and relationships based on peer pressure and pretending to be something you do not feel you really are to fit in.

Overall, peer identity was rated as the most important aspect of belonging for students from more-privileged social categories, including those who were boys, white and socio-economically advantaged. This may be connected to the findings that the schooling environment was frequently described as competitive and hierarchical; therefore, students from more-advantaged groups may find that friendship and peer status give them leverage to maintain a social advantage. For students who ranked peer identity as the most important aspect of belonging but experienced disadvantages such as having SEND, friendships were valued for their protective function.

### 3.4. Individual Identity Aspects of Belonging

#### 3.4.1. Quantitative Evidence from the Online Survey About the Importance of Individual Identity to Belonging

Indicators that referred to students’ individual identities were rated as the third most important constituent of belonging for the overall cohort of secondary-school-aged students in London.

There were three factors that spoke to students’ individual identities: ‘feeling confident to plan for my future,’ which was ranked fourth (mean 3.88); ‘feeling able to be myself in school,’ which ranked seventh (mean 3.67) and ‘having a say about decisions in school’, which ranked tenth (mean 3.48). While the survey ratings for social and cultural identity and peer identity were more proximally rated in their significance for students’ sense of belonging, the indicators for individual identity were relatively lower in their importance. While clearly an important aspect of children’s belonging, this suggests that the individual aspect of belonging was relatively less important than the social aspects previously discussed.

Similarly, the relative values of the individual identity indicators were somewhat more mixed among different groups. However, it was notable that socio-economically disadvantaged students rated all three identity factors relatively more highly than their peers, whereas students who were not socio-economically disadvantaged rated these factors slightly lower than the overall cohort.

Similarly, students with certain faith backgrounds ranked all three individual identity factors relatively higher than their peers, including students who were Muslim, Sikh, Hindu or Christian, while refugees ranked it as the second highest factor in their sense of belonging. Certain ethnic minority groups also ranked two of the three factors (including ‘feeling able to be myself in school’) as relatively higher than the overall cohort. This included children who identified as Black/Black British and Asian/Asian British.

It was also notable that white students ranked all three aspects of individual identity as relatively lower than the overall cohort, and boys ranked two of the three dimensions, including ‘feeling able to be myself in school’, as lower than the overall cohort, while girls ranked this factor as relatively more important than their peers.

#### 3.4.2. Qualitative Data on the Importance of Individual Identity

In explaining the importance of individual identity for belonging, students frequently cited the competitive and performative culture of schooling. One girl described her experience of the mainstream environment as ‘traumatising’ and explained, “I feel like a lot of the time in mainstream we’re on this path of becoming what we’re forced to be rather than who we are.” By contrast, students observed that when school creates the space and validation for developing non-academic aspects of students’ skills and interests, this generates a sense that one’s individual identity is supported:

“I think sort of the idea of like productive learning in the sense that like everything you do sort of works towards a goal that’s like beneficial, not just for like the school,—with results, -but for you as a person as well.” CW, Student, Secondary Mainstream B

“after school clubs where it builds on your mental and physical strength, for example boxing as I know boxing is one of the things that we do here at (PRU name) and quite a lot of students benefit from that because like it helps them not just physically but mentally as well because it gets them stronger and it teaches them life skills that they’re going to have for a long time…and that really builds someone as a person”NL, Student, Alternative Provision B

In a climate where schools, teachers and students are evaluated and funded according to narrow performance indicators, it may be that children’s sense of uniqueness and specialness regarding their individual skills and qualities can get easily quashed or minimised, especially for students from Black Caribbean backgrounds, as well as those from low socio-economic backgrounds, who are more likely to be excluded from mainstream education ([35]). Indeed, students from alternate provisions identified the celebration of individual identity as better served outside of mainstream schooling:

“I feel like (my current provision) does it better than most mainstream schools. They help you be who you are rather than what you’re meant to be or what you’re forced to become. I think in that sense, coming to (my current provision) especially made me realise that it’s not a bad place, but a place to start over again. I feel like a lot of time in mainstream we’re on this one path of becoming what we’re forced to be rather than who we are. So, in that sense, loving, they show us how to love ourselves and be who we are rather than what we’re meant to be.”MT, Student, Alternative Provision A

In reflecting upon why students from lower socio-economic backgrounds in particular value individual identity as an aspect of school belonging, we found it notable that these students observed the tendency to be judged both inside and outside of school, hence, the importance of feeling accepted for who you are:

“At school there is always like people who will be judging you for what you are doing… I feel like you’ll never feel that comfort at school.”EC, Student, Secondary Mainstream C (PP/FSM)

This may explain why, for children in less-advantaged groups, schools validating their individual identities was understood to be a key factor in developing self-confidence, which was seen to be of central importance in planning for the future and for well-being in the broader areas of social life:

“I feel like, if you are able to be yourself at school, that goes into being able to plan for your future because you need to be yourself before planning for the future.”DL, Student, Mainstream A (Muslim)

“If you’re not confident about your future like you’re not going to go anywhere. You need to be self-confident in yourself and what you’re going to do later on in life”NL, Student, Alternative Provision B

“I feel like the school should push for you to love who you are and just be confident with that even if that means you’re by yourself, you should still be happy without friends.”BC, Student, Secondary Mainstream C

It could be argued that, for non-advantaged groups, self-confidence and the freedom to express yourself are less achievable within friendship groups, which may explain why such students were at pains to distinguish between the value of friendships of different quality. Ultimately, individual identity may be seen to be conditional upon the ability to leverage social relationships to gain a meaningful sense of belonging in school:

“I feel like, once you know who you are as a person and you can be yourself, then other people will like you for who you are.”DL, Student, Secondary Mainstream, A

From examining these data on individual identity, we observed a close alignment between those groups who emphasised the relative importance of individual identity and those who identified the significance of social and cultural identity, namely, groups from less privileged and socio-economically advantaged backgrounds. The findings suggested that schooling climates (especially mainstream) were found to be particularly detrimental to self-worth and, therefore, confirmed the importance of measures that celebrated and catered to wider aspects of children’s development (Table 8).

## 4. Discussion

While cultural commentators have argued that young people are more isolated and lacking in real-world interaction today than in previous generations ([122]; [49]) due to the erosion of social connection in contemporary society ([96]), our findings suggest that young people in London see social identity (in its various forms) as the most important aspect to their sense of belonging in school. It is suggestive of societal structures as the root cause of this disconnectedness, rather than children’s preferences.

For London students as a whole cohort, cultural and social identity emerged as the most significant aspect of belonging, and more in-depth scrutiny of the data revealed that this aspect was of relatively greater significance for disadvantaged and underprivileged groups. In exploring why this may be the case, the qualitative findings indicated that respect for their cultural identities was a key feature of belonging, supporting other research with African American adolescents on the importance of respect for belonging, particularly within disadvantaged urban areas ([37]). Secondly, students identified the harmful effects of tokenistic attempts by the school to recognise cultural diversity. This builds upon findings from a study with refugee children who highlighted that when their cultures were discussed and represented in meaningful ways, coupled with an authentic interest in their homeland by peers and teachers, children felt a stronger sense of belonging and safety in school ([43]).

Finally, the data identified the importance of cultural identity in strengthening students’ sense of visibility in school, which corroborates research highlighting the psychic harm inflicted on students’ sense of connection with the school when their cultural identities are invisible ([33]). Underpinning these findings are the themes of understanding, recognition and safety that social and cultural identities can offer students, which may be especially important where children feel vulnerable to the dominance or aggression of key groups of students. Within this study, the protective status of social and cultural identity was reported particularly by girls and those from socio-economically disadvantaged backgrounds, for whom the positive impact has been identified as particularly important for their sense of self-esteem and social connectedness ([104]) and key to their mental health ([95]).

Peer identity was identified as a close second in terms of the most important factor in London school students’ sense of belonging and is well established for its impact on school belonging, acting as a motivational factor ([109]) as well as improving academic outcomes ([132]). Indeed, in this study, students across the board identified peer identity and friendship as the most important reason for coming to school. Regarding why peer identity emerged as being of relatively greater importance for more dominant groups such as boys, white students and those who are not socio-economically disadvantaged, one explanation is that by dint of their relatively more privileged status, where their cultural and social characteristics are more secure, visible and recognised, the peer identity factors are more salient to these dominant groups’ sense of belonging.

Relative to their disadvantaged peers, boys, white students and socio-economically advantaged children may feel that their cultural and social identities are less important to a sense of belonging due to their having less experience with marginalisation; therefore, their peer identities may be more significant by comparison. Concurrently, it could be argued that peer identity is more central to boys’ identities, given that boys’ friendships are more public, visible and secure, while girls’ friendships are more private, subject to frequent instability and conducted in the private and secluded parts of the school ([17]; [115]). Indeed, some research suggests that relationship losses are felt more intensely by girls, for whom close dyadic relationships are more important than for boys ([9]), indicating the relatively stronger sense of security boys enjoy in their friendships. The boys in this study also voiced the importance of peer identity for their sense of status, which, in line with other studies (e.g., [107]), could imply that there are greater social advantages to boys’ peer identities, which may even support their dominant status in schools. These findings underline the importance of a differentiated approach to examining belonging in schools that accounts for power structures within society, especially unequal power relations in terms of gender ([24]), social class ([52]) and race ([32]), as well as how these intersect. It is suggestive of solutions for policy and practice that also recognise these power inequalities.

As the third most important aspect of students’ sense of belonging for the young people of London, individual identity was highlighted as a key mechanism for buffering the performative culture of schooling. The negative impact of standardised assessment indicators on children’s schooling identities has been well documented ([97]; [116]). Research that has explored ameliorative measures aiming to support aspects of individual identity such as personal interests, agency and individuality has had a beneficial impact on children’s sense of belonging in school ([99]). In considering why individual identity may be particularly important for the less advantaged groups, who rated it as being relatively more significant compared to their more privileged peers, there is some research that indicates that individual identity is used by marginalised groups as a coping strategy in contexts such as schools where children’s cultural identities are not validated ([90]).

Furthermore, schools’ support of individual identity through agency, individual interests and individual strengths has been found to promote self-confidence and, in turn, more-positive future aspirations and motivation ([98]; [121]). Finally, it may be that students from less advantaged backgrounds have more ‘labour’ to undertake in ‘fitting in,’ such that being ‘authentic’ is a less available personal choice. This would echo recent findings that suggest that university students from high-status groups that align with the university prototype more readily see themselves as fitting in without ‘strain or dissonance.’ Less advantaged students, however, may prioritise fitting in because authenticity may feel comparatively less achievable from an identity perspective ([46]). For disadvantaged groups, therefore, ‘a sense of belonging and the feeling of connection with others might constrain their sense of authenticity’ ([46]). Extending the reach of these studies, our findings suggested that the self-confidence generated from feeling affirmed in school led students to feel more positively about their futures as well as their present experiences in school. This echoes [44]’ ([44]) argument that personal and collective identities impact motivated action, which would include school engagement and future aspirations.

This leads us to consider the interconnection between social and individual aspects of students’ identities in school and how they relate to belonging. Early work, primarily within Social Identity Theory, has tended to overlook personal identity in emphasising the importance of social aspects of identity in our drive to achieve a sense of belonging ([120]). Later developments within the tradition have acknowledged that social and individual identity are distinct categories, context-dependent, where individual identity is mobilised largely through the concept of motivation ([63]). Identity Process Theory can be seen as an attempt to assert the importance of individual identity in recognising that individuals have a need to maintain a sense of coherence in negotiating between their various social identities and, as such, perceive a sense of threat where there is conflict between the different social demands and memberships ([14], [15]). This marked a shift towards centring individual agency in identity formation through the notion of an unrelenting agentic drive to maintain a coherent thread between the different social relationships that influence our lives. In broadening the field from the study of group-level categorisations to individually situated meaning-making, SIT scholars have moved towards the concept of narrative identity advanced through post-structuralist traditions ([62]; [79]). This position construes the self as fluid, multiple and contextual, where the focus shifts from the social structure of identity constituents towards the notion of identity as a process where the most important aspect is the stories of self we tell ourselves ([113]). This echoes [54]’ ([54]) notion of an individual’s ‘narrative of self’, which he posited was fundamental to achieving their ‘ontological security’. In this sense, identity is not just a reflection of our experiences and sense-making; the stories of self also have a productive power in shaping what is meant by belonging, for example, in schools.

Positive psychology also aligns with the perspective that identity is malleable and argues that schools should play a key role in enabling children to develop the individual characteristics or ‘virtues’ that enable prosocial behaviour, belonging and well-being ([111], [112]). However, by reducing the social to the impact of positive relationships on account of desired individual identity qualities, and through arguing that these characteristics are universal, it misses a consideration of cultural context and power dynamics.

By contrast, social psychologists have taken more account of the power inequalities within identity positions in seeking to understand how the individual navigates between multiple social identity positions. When applying an intersectional lens, it has been argued that social identities are not simply additive categories (such as gender, ethnicity and social class), but rather they interact with each other, and the salience of these categories is a consequence of the societal power structures that they operate within. Here, further research could usefully take an intersectional lens in exploring the relative importance of social and individual forms of identity for belonging in school to develop solutions for policy and practice that are more closely reflective of young people’s positioning within prevailing power structures.

More recently, Social Identity Theory has evolved to advocate more integration between the social and individual aspects of our identity, arguing for the need to balance the seemingly opposing needs of social membership with individual distinctiveness ([64]; [57]). Nevertheless, in seeking this task as a quest for equilibrium, the argument misses the ways in which social and individual identity are co-constitutive in defining each other in dynamic and inseparable ways. The findings from this study advance an understanding of the interconnection between social and individual identity in belonging in highlighting that a secure sense of social identity is a prerequisite for generating a valued self-identity. It may be that, for marginalised groups, cultural identities are more important social identities, while for more-privileged groups, friendships are more significant. However, for students to be able to mobilise their social identities to generate a secure sense of belonging in school, they are required to strengthen and affirm individual identities so as to ‘be themselves’ in ways that go beyond prescriptive and performative iterations of what it means to be an ideal student (see [68]) by instead celebrating the whole child. This is something that may be especially important for children from marginalised or disadvantaged groups, who are feasibly further from this ideal ([136]). As such, while they are conceptually distinct categories, social and individual identity aspects of belonging are self-reinforcing and mutually co-constitutive. This points to the importance of schooling strategies to belonging being based on relational and identity-based ideals that position the unique and individual child at the heart of their approaches, as can be seen, for example, in recent approaches emphasising a collectivist and collaborative as opposed to individualistic well-being approach (see [22]).

With higher levels of ‘all aspects of school belonging’ associated with ‘lower mental health symptoms across adulthood,’ ([5]), it is vital that schools are supported to consider and support a sense of school belonging for all students. Given that the rates of mental health difficulties are particularly high among specific groups, including young people with minority gender identities ([123]), those who identify as LGBTQ+, those from the poorest 20% of households, those having a learning disability, those with autism, students who are deaf, those from African-Caribbean communities ([28]) and students who are care-experienced ([93]), a nuanced understanding of school belonging that pays attention to the identity and relational processes of different social groups’ needs is greatly needed. If a salutogenic, preventative approach to the mental health crisis facing young people is to genuinely meet the needs of the whole student body, then further attention to these three identity aspects of belonging—social and cultural identity, peer identity and individual identity—both in research and targeting support, is key. This warrants more extensive longitudinal work to build more rigorous data to understand the role belonging and identity plays in mental health promotion.

### Limitations and Future Research

This study has several limitations that should be acknowledged. First, this research was based on a single metropolitan context, drawing on survey responses from 2078 secondary-aged students and focus groups with 76 students in Greater London. While the diversity of the city offers rich insights into the role of social, peer and individual identities in school belonging, the findings may not fully reflect experiences in rural areas, smaller towns or other national contexts. Second, the cross-sectional design limits the ability to establish causal relationships between identity, belonging and well-being. Longitudinal studies would be valuable to track how belonging and identity processes evolve across developmental stages and shifting educational contexts. Third, although qualitative data provided depth to the quantitative findings, the relatively small number of focus group participants restricts the range of experiences that could be captured, particularly among minority or intersectional-identity groups. Future research could extend these findings in several directions. In particular, there is a need to investigate how belonging operates beyond the school doors, given that young people’s social identities are also shaped by their families, communities, neighbourhoods and online environments. Understanding how school-based belonging interacts with broader social belonging may be particularly important for marginalised groups, whose identities may be either affirmed or challenged outside school settings. Expanding research across multiple geographical and cultural contexts and employing longitudinal and mixed-methods designs would strengthen the evidence base. In addition, future work should explore interventions that explicitly connect school belonging to wider social and civic forms of belonging, recognising that young people’s sense of identity extends beyond the institution of schooling and into the social worlds that frame their everyday lives.

## 5. Conclusions

This study set out to explore how different dimensions of identity shape young people’s sense of belonging in the case of secondary-school-aged children in London. The findings highlight that social and cultural, peer and individual identities each play distinct yet interconnected roles, with their relative importance varying across student groups. From this work emerges a conceptual model of the three identity dimensions of belonging, their interconnections and their implications for well-being (see Figure 1).

While conceptually distinct, these identity aspects are mutually reinforcing and co-constitutive: a secure social identity supports the development of valued individual identities, while peer relationships mediate belonging across groups. Belonging, in turn, is linked to improved well-being in school, which underlines the importance of relational and identity-based approaches to school practice and policy. These insights align with growing public and policy recognition of the importance of belonging, supporting a wider shift away from individualising well-being approaches and towards collaborative strategies that build and affirm students’ identities and relationships. In demonstrating the interplay between identity and belonging, this study contributes to the field by offering an empirically grounded framework that not only explains the variation in students’ experiences of belonging but also provides a foundation for interventions that recognise the whole child within their social worlds.

## Figures and Tables

**Figure 1 behavsci-15-01421-f001:**
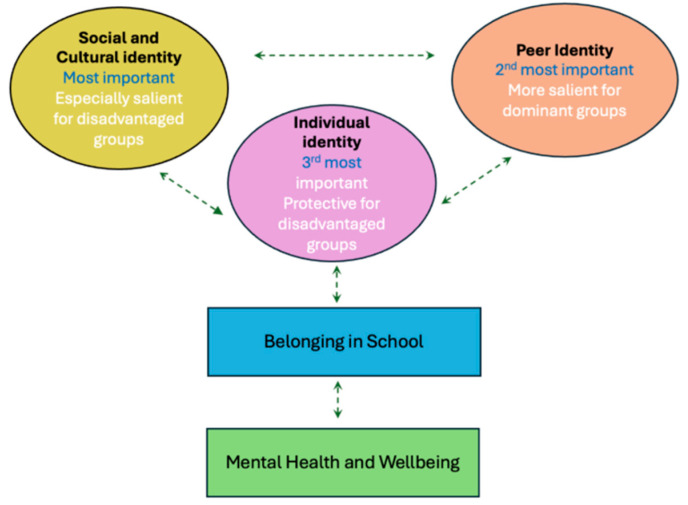
A conceptual model of the three identity dimensions of belonging (social and cultural, peer, individual), their relative importance for different student groups and their interconnections.

**Table 1 behavsci-15-01421-t001:** Overview of survey items and sources.

Survey Item “*To feel a sense of belonging, how important is…*”	Sources
Having someone I trust to talk to at school if I have a problem	[82] ([82]); [119] ([119]); [127] ([127]); [55] ([55]); [34] ([34])
Positive relationships between staff and students	[4] ([4]); [45] ([45]); [74] ([74]); [5] ([5]); [127] ([127]); [117] ([117]); [1] ([1])
Having a say in decisions about the school	[40] ([40]); [101] ([101]); [48] ([48])
Taking part in activities, clubs, trips and events at school	[82] ([82]); [61] ([61]); [2] ([2]); [55] ([55]); [60] ([60])
People noticing when I am good at something	[55] ([55]); [12] ([12])
Getting along with other students at school	[45] ([45]); [4] ([4]); [127] ([127]); [27] ([27])
Having a group of friends in school that I trust	[82] ([82]); [27] ([27]); [61] ([61]); [1] ([1])
Families/parents/carers feeling welcome to get involved with what goes on in school	[45] ([45]); [82] ([82]); [127] ([127]); [102] ([102])
People from all backgrounds (e.g., ethnicities, family income levels, sexualities) feeling welcome and heard in school	[124] ([124]); [117] ([117]); [92] ([92]); [78] ([78]); [1] ([1]); [34] ([34])
Having a space where my friends and I can hang out in school	[81] ([81]); [101] ([101]); [17] ([17])
Seeing different backgrounds (e.g., culture, religions, races, etc.) represented in school displays and celebrations	[16] ([16]); [29] ([29]); [130] ([130]); [102] ([102]); [34] ([34]); [84] ([84]); [25] ([25]); [39] ([39])
Being treated with as much respect as everyone else at school	[4] ([4]); [3] ([3]); [2] ([2]); [114] ([114]); [55] ([55])
Talking about important things that are happening in the world (e.g., Climate Change, wars, Black Lives Matter, Me Too)	[72] ([72]); [108] ([108]); [59] ([59]); [71] ([71]); [77] ([77]); [50] ([50]); [76] ([76])
Having the chance to get involved and make a difference in other people’s lives	[59] ([59]); [2] ([2]); [80] ([80]); [53] ([53]); [75] ([75]); [50] ([50])

**Table 2 behavsci-15-01421-t002:** Overview of covariates collected and linked to the survey.

Category	Variables Collected	Source of Data
Demographics	Gender; Sexuality; Faith;	Self-report (survey)
Educational needs	Disability/SEND; Neurodivergence/English as an Additional Language (EAL)	Self-report (survey)
Care experience	Care experience; Young carer	Self-report (survey)
Migration status	Refugee; Asylum seeker;	Self-report (survey)
Socio-economic status	Free School Meal eligibility; Pupil Premium	Linked anonymised administrative data (school record)

**Table 3 behavsci-15-01421-t003:** Secondary-aged students’ ratings of factors impacting their school belonging.

Factor	Whole Cohort Mean	Rated
Being treated with as much respect as everyone else at school.	4.26	1
Having a friend or group of friends in school that I trust.	4.25	2
People from all backgrounds (e.g., ethnicities, family income levels, sexualities) feeling welcome and heard in school.	4.01	3
Feeling confident to plan for my future.	3.88	4
Getting along with other students at school.	3.82	5
Seeing different backgrounds (e.g., cultures, religions, races, etc.) represented in school displays and celebrations.	3.76	6
Feeling able to be myself at school.	3.67	7
Having a space where my friends and I can hang out in school.	3.64	8
Positive relationships between staff and students.	3.57	9
Having a say in decisions about the school.	3.48	10
Talking about important things that are happening in the world (e.g., Climate Change, wars, Black Lives Matter, Me Too).	3.48	11
Having someone I trust to talk to at school if I have a problem.	3.36	12
People noticing when I am good at something.	3.34	13
Having the chance to get involved and make a difference in other people’s lives.	3.29	14
Taking part in activities, clubs, trips and events at school.	3.09	15
Families/parents/carers feeling welcome to get involved with what goes on in school.	2.77	16
My school having connections to the local community.	2.63	17

**Table 4 behavsci-15-01421-t004:** Aspects of identity referred to by secondary-aged students’ top ten rated factors impacting their sense of school belonging.

Rated	Factor	Identity Aspect
1	Being treated with as much respect as everyone else at school.	Social and cultural identity
2	Having a friend or group of friends in school that I trust.	Peer identity
3	People from all backgrounds (e.g., ethnicities, family income levels, sexualities) feeling welcome and heard in school.	Social and cultural identity
4	Feeling confident to plan for my future.	Individual identity
5	Getting along with other students at school.	Peer identity
6	Seeing different backgrounds (e.g., cultures, religions, races, etc.) represented in school displays and celebrations.	Social and cultural identity
7	Feeling able to be myself at school.	Individual identity
8	Having a space where my friends and I can hang out in school.	Peer identity
9	Positive relationships between staff and students.	Relationships
10	Having a say in decisions about the school.	Individual identity

**Table 5 behavsci-15-01421-t005:** Identity components and their link to belonging in school.

Identity Component	Explanation	Link to Belonging
Social and Cultural Identities	Shaped by wider social structures and cultural contexts such as ethnicity, gender, language, religion and socio-economic background. They provide a sense of belonging to broader communities (including the family) and influence how students are recognised within the school environment.	Belonging is strengthened when schools affirm and value students’ cultural and social identities; it is undermined when these identities are marginalised or rendered invisible.
Peer Identities	Formed through friendships, peer groups and classroom dynamics, these identities are central to how children experience acceptance, inclusion or marginalisation. Peer identities often mediate feelings of belonging in day-to-day school life and can buffer or intensify the impact of wider social disadvantage.	Positive peer recognition and supportive friendships foster belonging, whereas exclusion, bullying or peer conflict erode it.
Individual Identities	Reflect personal attributes, values, aspirations and self-concept. Individual identity shapes how children interpret their experiences and negotiate between cultural expectations, peer pressures and personal goals, providing a sense of agency and uniqueness within the school setting.	Belonging is enhanced when schools allow space for individual expression and agency, enabling children to feel seen and valued as unique persons.

**Table 6 behavsci-15-01421-t006:** Relative importance of social identity factors for different groups by mean and ranking.

Social Identity Factor	Whole Cohort Mean	Whole Cohort Mean Position	Groups for Whom Factor Was Particularly Important Mean Rating Position ^1^	Groups for Whom Factor Was Relatively Less Important Mean Rating Position ^1^
“Being treated with as much respect as everyone else in the school”	4.26	1st	Low SES/deprived (4.34) *1st Girls (4.44) *1st Cisgender (4.36) *2nd Christian (4.34) *1st Sikh (4.43) *1st Muslim (4.32) *1st Black/Black British (4.27) *1st Asian/Asian British (4.26) *1st EAL (4.27) *1st Neurodivergent (4.21) *1st	Not deprived (4.25) *2nd Boys (4.08) *2nd Gender minorities (3.74>) Transgender (3.5) *2nd to 4th All other sexual minority identities (3.77>) *2nd Jewish (3.63) Buddhist (3.83) Hindu (4.2) *2nd–3rd No religion (4.23) Atheist (4.18) SEND (3.79) *3rd Refugee (3.5) Asylum seeker (3.36) *2nd In care (3.71) *3rd White (4.27) *2nd
“People from all backgrounds feeling welcome and heard in school”	4.01	3rd	Low SES/deprived(4.08) *3rd Girls (4.28) *3rd Cisgender (4.36) *3rd Hindu (4.38) *1st No religion (4.03) *3rd Atheists (4.08) *4th White (4.02) *3rd Asian/Asian British (4.06) *3rd EAL (4.05) *3rd Neurodivergent (3.96) *3rd	Not deprived (4) *3rd Boys (3.75) *5th Transgender (3.5) and all other gender mins (3.74>) *9–11th All sexual minorities (except bi) 3.42>) *7th–10th Black/Black British (3.96) *4th Muslims (3.96) *4th Young carer (3.92) *3rd In care (3.6) *4th SEND (3.44) *5th Refugee (3.31) *6th Asylum seeker (2.97) *7th
“Seeing different backgrounds (e.g., cultures, religions, races, etc.) represented in school displays and celebrations”	3.75	6th	Low SES/deprived (3.94) *5th Girls (4.04) *4th Bisexual (3.84) *6th Christian (3.79) *5th Muslim (3.82) *5th Hindu (4.08) *4th Sikh (4) Black/Black Brit (3.88) *5th Asian/Brit Asian (3.86) *5th EAL (3.88) *5th	NOT deprived (3.75) *6th Boys (3.51) *6th All gender mins (3.17>) *7th–11th All other sex mins (3.36>) *6th–9th No religion (3.72) *6th White (3.73) *6th Neurodivergent (3.64) *6th SEND (3.39) *8th Refugee (3.33) *5th Asylum seekers (3.18) *5th In care (3.57) *6th

^1^ Items in black indicate that the mean is higher than that for the whole cohort. Items in red indicate that the mean is lower than that for the whole cohort. Items in blue indicate a more subjective rating where the mean is higher/lower than average but the ranking of the factor reveals a different view, clarifying the relative importance of the factor for the cohort. For each group, the identity factors were ordered by mean score, from the highest mean score to the lowest mean score. The mean rating position refers to where the item appears in this organisation.

**Table 7 behavsci-15-01421-t007:** Relative importance of peer identity factor for different groups by mean rating and ranking.

Peer Identity Factor	Whole Cohort Mean	Whole Cohort Mean Position	Groups for Whom Factor Was Particularly Important Mean Rating Position ^1^	Groups for Whom Factor Was Relatively Less Important Mean Rating Position ^1^
“Having a friend or group of friends in school that I trust”	4.25	2nd	Not deprived (4.28) *1st White (4.34) *1st Girls (4.36) *2nd Heterosexual (4.28) *1st Cisgender (4.26) *2nd Young carers (4.34) *1st No religion (4.33) Boys (4.19) *1st In care (3.95) *1st Refugee (4.04) *1st Asylum seeker (3.72) *1st SEND (3.97) *1st Atheists (4.23) *1st Jewish (4.16) *1st All sexual minorities (3.98>) *1st All gender minorities (3.88>) (except Transgender) *1st	Asian/Asian British (4.22) *2nd Black/Black British (4.1) *2nd Muslims (4.21) *2nd Neurodivergent (4.21) *2nd
“Getting along with other students at school.”	3.82	5th	Low SES/deprived (3.86) *6th Not deprived (3.84) *5th Cisgender (3.83) *5th Heterosexual (3.94) *4th White (3.86) *4th Christian (3.84) *5th Hindu (3.87) *6th No religion (3.87) *4th Young carers (3.85) *4th SEND (4.04) *1st Boys (3.77) *4th Transgender (3.76) *1st Non-binary (3.81) *1st Asylum seeker (3.5) *3rd	In care (3.73) *5th Sexual minorities (3.58>) *2nd–9th Black/Black British (3.73) *6th Asian/Asian British (3.81) *5th Muslim (3.77) *6th Girls (3.92) *6th Not deprived (3.84) *6th
“Having a space where me and my friends can hang out in school”	3.64	8th	Low SES/deprived (3.76) *8th Girls (3.75) *8th Cisgender (3.72) *9th Heterosexual (3.79) *6th Bisexual (3.83) *7th White (3.67) *7th Christian (3.73) *7th Hindu (3.8) *8th Boys (3.56) *5th Gay/Lesbian (3.59) *3rd Not deprived (3.64) *7th	All gender minorities (3.28>) *6th–8th EAL (3.62) *8th Neurodivergent (3.6) *8th SEND (3.3) *9th Refugee (3.26) *8th Asylum seekers (2.82) *10th Black/Black British (3.59) *8th No religion (3.54) *9th Atheist (3.58) *8th Muslim (3.63) *8th Sikh (3.47) *9th Buddhist (3.39) *6th Jewish (3.47) *5th

^1^ Items in black indicate that the mean is higher than that for the whole cohort. Items in red indicate that the mean is lower than that for the whole cohort. Items in blue indicate a more subjective rating where the mean is higher/lower than average but the ranking of the factor reveals a different view, clarifying the relative importance of the factor for the cohort. For each group, the identity factors were ordered by mean score, from the highest mean score to the lowest mean score. The mean rating position refers to where the item appears in this organisation.

**Table 8 behavsci-15-01421-t008:** Individual identity factors by the groups that rated them more or less important.

Individual Identity Factor	Whole Cohort Mean	Whole Cohort Mean Position	Groups for Whom Factor Was Particularly Important Mean Rating Position ^1^	Groups for Whom Factor Was Relatively Less Important Mean Rating Position ^1^
“Feeling confident to plan for my future”	3.88	4th	SES Disadvantaged (4.03) *4th Heterosexual (3.91) *5th Black/Black British (4.02) *3rd Asian/Asian British (3.91) *4th Buddhist (4.06) *1st Hindu (4.07) *5th Muslim (4.06) *3rd Christian (3.95) *4th EAL (4) *4th Boys (3.87) *3rd Asylum seeker (3.52) *2nd	Not deprived (3.86) *4th White (3.82) *5th In care (3.63) *4th SEND (3.52) *4th Refugee (3.45)* 4th Neurodivergent (3.72) *5th All gender mins (3.41>) *5th–8th All sexual mins (3.55>) *5th–9th Girls (3.93) *5th
“Feeling able to be myself in school”	3.67	7th	SES Disadvantaged (3.86) *7th Girls (3.88) *7th Heterosexual (3.68) *8th Bisexual (3.88) *5th Black/Black British (3.8) *6th Mixed heritage (3.69) *6th Christian (3.73) *7th Muslim (3.68)* 7th Sikh (3.86) *5th Hindu (3.85) *7th Transgender (3.32) *4th Non-binary (3.41) *5th	Not deprived (3.64) *7th/8th Boys (3.49) *8th White (3.65) *8th Young carer (3.62) *9th SEND (3.44) * 6th Refugee (3.12) *9th Asylum seeker (2.74) *12th In care (3.28) *9th Other sexual mins (3.41>) *6th–8th Jewish (3.11) *11th Buddhist (3.33) *10th
“Having a say about decisions about the school”	3.48	10th	Low SES/deprived(3.63) *10th Cisgender (3.6) *10th Christian (3.58) *10th Muslim (3.6) *9th Sikh (3.61) *6th Hindu (3.67) *10th Neurodivergent (3.49) *10th EAL (3.56) *10th Transgender (3.35)* 5th Other gender mins (3.18>) *7th/8th Asylum seekers (3.17) *8th In care (3.39) *7th	Not deprived (3.47) *10th Boys (3.43) *10th White (3.41) *12th SEND (3.04) *11th Refugee (3.09) *11th Girls (3.66) *11th

^1^ Items in black indicate that the mean is higher than that for the whole cohort. Items in red indicate that the mean is lower than that for the whole cohort. Items in blue indicate a more subjective rating where the mean is higher/lower than average but the ranking of the factor reveals a different view, clarifying the relative importance of the factor for the cohort. For each group, the identity factors were ordered by mean score, from the highest mean score to the lowest mean score. The mean rating position refers to where the item appears in this organisation.

## Data Availability

The data presented in this study are available on request from the corresponding author.

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
