# Peer review of "Resilience Through Belonging: Schools’ Role in Promoting the Mental Health and Well-Being of Children and Young People"

_behavsci, 2025, doi:10.3390/bs15101421_

Round 1
Reviewer 1 Report
Comments and Suggestions for Authors
Thank you for conducting this study, as it seems to move the field forward in considering a more balanced approach to addressing student well-being that is centered on their personal and collective identities.
The abstract and introduction are well written and I have very few comments here. First, I keep checking the title and adding a possessive to school (i.e., Schools’ role) and then taking it away. I would ask the authors to discuss this as adding it would be consistent with the first sentence and other sentences throughout the paper. There are a few long sentences that can be revised for clarity for example: (1) lines 13-17 of the abstract, (2) lines 72-77 which is important to clarify since this is defining a key term of the study. In line 132, you refer to the importance of an identity-based approach - say more about this as a collective/relational approach. In this same paragraph (lines 135-136), I am not sure this sentence is relevant to the point of the paragraph. If you disagree, please explain how in the text. Line 161, you refer to “well” and I believe this is supposed to be well-being. You will need to check this.
The biggest area of improvement I have identified for this manuscript is its methodology, not the use of a sequential mixed methods approach, but the lack of mention of the specific model and the approach to reporting the results. I encourage you to specifically state/name your mixed methods approach. This can be easily incorporated in the first and third paragraphs of your methodology. As I read it now, you conducted the survey (quantitative with an open ended question) and these results informed your focus groups. This could be described as a Sequential Explanatory Design (Ivankova, Creswell, & Stick, 2006) or similar design. As such, the results should follow this format as well, with an integration of the phases at the end of the analysis.
Additional points regarding the survey items - (1) since you have compiled the scale from a literature review, a table with sources of your survey items is appropriate; (2) a table with your covariate information can be provided to the reader (lines 215-221 of methodology).
This would change how the results are reported by revising Section 3 such that you report the results from phase 1 first, followed by phase 2, and then integrating these phases. Table 1 described all 17 items and the Identity aspects (as indicated by your literature review) can be an added column. You will need a note to indicate the rating scale for your measure. The end of these results should identify your top 10 issues across the whole cohort (lines 225-226).
Next, you should report the open-ended response data from Phase 1.
Following your method (lines 226-228), Table 3 should indicate the relative importance of social identity factors for different groups for each of the top 10 items.
Based on lines 228-229, your next section of results should reflect the focus groups conducted. Again, an integrative interpretation can be provided once these results have been clearly laid out. It seems your focus groups revealed a third identity (peer identity) that is an important finding and perhaps your integrative section can emphasize it utilizing the data that you present in 3.2.
Your discussion is well written. The only point that I struggled to find goes back to lines 178-187, particularly the idea of belonging beyond school doors as important to one’s social identity. It is unfortunate that the items that directly relate to this were not in the whole cohort’s top 10 ratings as this could be valuable for focus group students to consider how these aspects could be important to their sense of belonging. Perhaps this is not a finding in your current study but something that can be considered in future ones. Lastly, I would be remiss if I did not share an article (Eccles, 2009) that considers personal and collective identities as motivators of action.
Overall, this manuscript is of high quality. The recommendations above add clarity and structure for your audience, as the substance of your work is well done.
Author Response
Comment 1: Thank you for conducting this study, as it seems to move the field forward in considering a more balanced approach to addressing student well-being that is centered on their personal and collective identities.
Thank you for your helpful review
Comment 2:
The abstract and introduction are well written and I have very few comments here. First, I keep checking the title and adding a possessive to school (i.e., Schools’ role) and then taking it away. I would ask the authors to discuss this as adding it would be consistent with the first sentence and other sentences throughout the paper. There are a few long sentences that can be revised for clarity for example: (1) lines 13-17 of the abstract, (2) lines 72-77 which is important to clarify since this is defining a key term of the study. In line 132, you refer to the importance of an identity-based approach - say more about this as a collective/relational approach. In this same paragraph (lines 135-136), I am not sure this sentence is relevant to the point of the paragraph. If you disagree, please explain how in the text. Line 161, you refer to “well” and I believe this is supposed to be well-being. You will need to check this.
Response: Thank you for flagging the inconsistency around apostrophe usage. We have decided that our point refers to the role of schools in general, so have used the multiple form. We have broken down the longer sentence in the abstract (13-17) to read: Our social identities are multiple and individual aspects of our identities are multilayered. A more nuanced consideration of children’s sense of belonging across the different social domains of their lives is therefore important in developing wellbeing approaches which prevent poor mental health outcomes for all children. We have also reworded for clarity: "Three distinct theoretical positions are notable among school-based mental health approaches that have sought to strengthen children’s mental health and wellbeing (see Howdle Lang, 2022 for a review). " We have also expanded this paragraph to define more clearly the different approaches in order to try to clarify the term of our study as you suggest. We have highlighted the whole section in yellow to highlight where we have amended.
Comment 3:
The biggest area of improvement I have identified for this manuscript is its methodology, not the use of a sequential mixed methods approach, but the lack of mention of the specific model and the approach to reporting the results. I encourage you to specifically state/name your mixed methods approach. This can be easily incorporated in the first and third paragraphs of your methodology. As I read it now, you conducted the survey (quantitative with an open ended question) and these results informed your focus groups. This could be described as a Sequential Explanatory Design (Ivankova, Creswell, & Stick, 2006) or similar design. As such, the results should follow this format as well, with an integration of the phases at the end of the analysis.
Thank you for your helpful feedback and suggestion.
We have included the mention of a sequential explanatory design.
We have also inserted text explaining the theoretical underpinnings to our mixed method research design, detailing our alignment with Haig and Evers’ critical realist approach.
As integration of the phases happened during the analysis, we have amended our explanation to make that clearer and reworked the methodology to try to describe it more fully for the reader.
We have clarified the sequential approach through the subheadings to try to more clearly delineate between the quantitaive data and qualitative data.
Comment 4 : Additional points regarding the survey items - (1) since you have compiled the scale from a literature review, a table with sources of your survey items is appropriate; (2) a table with your covariate information can be provided to the reader (lines 215-221 of methodology).
Thank you for these suggestions. We have now included both tables suggested. Please see Table 1 detailing sources from the literature review which informed the different survey items.. Please see Table 2 for details of the different variables collected through the survey itself and associated anonymised data provided by the participating schools.
Comment 5:
This would change how the results are reported by revising Section 3 such that you report the results from phase 1 first, followed by phase 2, and then integrating these phases. Table 1 described all 17 items and the Identity aspects (as indicated by your literature review) can be an added column. You will need a note to indicate the rating scale for your measure. The end of these results should identify your top 10 issues across the whole cohort (lines 225-226).
Clarification has now been provided on why we presented the analyses and results in the way we did, following the approach adopted by Haig and Evers which is distinct from an integrative approach wherein interpretation rests at the point of integration. We have therefore not restructured the presentation of the results but have aimed to signpost the two phases more clearly through the subheadings. We added information as you recommend about the rating scale used in the measure: "This asked them to rate the importance of 17 belonging related factors. The rating scale for each item was 0 (not important), 1 (slightly important), 2 (quite important), 3 (important), 4 (very important) and 5 (extremely important). "
Comment 6: Next, you should report the open-ended response data from Phase 1.
We would like to apologise for this error on our part. This student survey did not infact have an open-ended response option. A similar survey we were working on does have and we mistakenly included refernce to it in writing up the methodology. We have now removed mention of this as it is incorrect.
Comment 7: Following your method (lines 226-228), Table 3 should indicate the relative importance of social identity factors for different groups for each of the top 10 items.
Thank you. Sorry for this oversight. We have now included this information within this table - and in some of the other tables where we had missed it.
Comment 8: Based on lines 228-229, your next section of results should reflect the focus groups conducted. Again, an integrative interpretation can be provided once these results have been clearly laid out. It seems your focus groups revealed a third identity (peer identity) that is an important finding and perhaps your integrative section can emphasize it utilizing the data that you present in 3.2.
Response: Peer identity did not emerge from the focus groups, but rather was always part of the theoretical framework which we now discuss at the outset of the paper.
Comment 9: Your discussion is well written. The only point that I struggled to find goes back to lines 178-187, particularly the idea of belonging beyond school doors as important to one’s social identity. It is unfortunate that the items that directly relate to this were not in the whole cohort’s top 10 ratings as this could be valuable for focus group students to consider how these aspects could be important to their sense of belonging. Perhaps this is not a finding in your current study but something that can be considered in future ones. Lastly, I would be remiss if I did not share an article (Eccles, 2009) that considers personal and collective identities as motivators of action.
Response. Thank you for the useful introduction to Eccle's paper. We have built reference to this into the LIterature Review and discussion. As you identify the findings discussed here mean that some aspects of collective identity beyond the school gates are not really explored. We have built this into a Limitations section to flag that this is an important area that would benefit from further exploration.
Comment 9: Overall, this manuscript is of high quality. The recommendations above add clarity and structure for your audience, as the substance of your work is well done.
Thank you for your thorough and helpful review which has been extremely helpful in our efforts to sharpen the clarity of our article for the reader.
Reviewer 2 Report
Comments and Suggestions for Authors
Thank you for the opportunity to review your manuscript. The authors begin with an introduction about student wellbeing with an emphasis on relevant literature in England, as well as information about individual resilience in the literature. This manuscript has great potential, and the topic of student wellbeing is one that remains at the forefront of education research around the world. They provide nice connections with relevant literature and provide a brief overview of the Methodology.
More clarity can be provided throughout, especially regarding the data analysis. Greater clarity of themes within the results section would make their argument clearer. There is a lot of information that can get lost in this section. The authors should add separate sub-sections for limitations and future research, as well as a conclusion section to briefly summarize the work, why this work is important, and how it benefits the field. I would recommend that the authors review formatting for in-text citations to be consistent throughout and clarity of language. I have noted some specific recommendations for the authors below. While I did note a lot of recommendations and areas to consider revising, I think this manuscript is worth considering after major revisions. I see great potential, after more focused revisions, in how it can contribute to the field on student wellbeing.
Recommendations for Authors:
- Introduction-
- You discuss changes based on the pandemic, but many of the references you cite are pre-pandemic. I would recommend making sure to connect claims about shifts in how student wellbeing is viewed to post pandemic literature to strengthen that argument.
- Page 2, line 41
- Review formatting of citations
- Page 2, line 57
- Unclear why you used a footnote to cite the quote instead of a citation to stay consistent with journal formatting.
- Same comment for quote on page 4, lines 142-143.
- Please make sure you are adding a page number for direct quotes. I would check this throughout your manuscript.
- Page 2, lines 68-69
- What do you mean by “individualizing” and “moralizing”? I would expand on this statement more.
- Page 3, Line 86
- Be mindful of spacing
- Page 3, line 107
- Review grammar
- Page 3, lines 115-116
- You state “developed by” in your sentence and then add parentheses around the people you are mentioning. I would take their name out of parentheses and leave the year in parentheses.
- In-text citations
- Please review journal policy for how to format citations with multiple authors to ensure you are consistent in the use of et al. and when to include the last names of all authors in your citation.
- With APA 7th edition you would use et al. for references with three or more authors.
- Page 3, Lines 116-117
- What school interventions are you referring to in this sentence?
- Page 3, lines 131-135
- Please add citations to support this information.
- Page 4, lines 144-151
- I would add some examples of the principles/findings.
- Page 5, line 161-162
- Please consider revising this definition for clarity. Making sure your readers understand how you define belonging is important. I am not sure what you mean by embracing the “social and well” in this sentence. The introduction before the quote needs greater clarity.
- Page 4, line 176
- Delete period before in-text citation
- Page 4 & 5, Lines 179-187
- Consider separating this sentence into multiple sentences.
- Page 5, line 186
- Be mindful of spacing within citation
- Page 5, line 216
- Please review word choice
- Page 5, line 218
- What does SEND stand for? Want to make sure this introduced for an international readership that my not use this acronym.
- Methodology
- Participants and Setting
- Please include sub-sections where you provide information about the participants and the setting in greater detail. You provide brief bits of this information in your overall Methodology section, but please provide this information in greater detail.
- Knowing more about the participant demographics is important, especially because you make claims about students from “non-dominant groups” in your results. The reader needs to know more about the demographics of participants to support claims made in the results.
- Data Analysis
- Please add a subsection explaining the data analysis used for both quantitative and qualitative data. This needs to be made much clearer to support replication and to inform the results.
- Page 6, Lines 239-240
- In a research manuscript the heading should be Results. What you have would be a relevant sub-heading for the results section.
- Consider adding additional sub-headings within your Results section to highlight specific themes identified. You have a lot of information and the main points get lost in the this section.
- Identity
- I would consider adding a table where you identify the identity aspects you are focusing on and how you define them. This would be a beneficial reference point for readers to refer to as you describe the results and discussion section. For example, this would be a nice accompaniment for Table 2 where you align factors with identity aspects.
- Repeat same paragraph twice.
- Page 6, Lines 246-251
- Page 7, Lines 254-259
- Review Results section for content that would be more appropriate for the discussion section, such as recommendations for future research.
- Page 8, lines 294-295
- Page 9, line 302-303
- Have a sub-heading but no text description.
- Section 3.1.3
- Consider adding sub-headings based on the three aspects that were identified in the qualitative data.
- It is unclear from the current methods section how this data was analyzed, so you want to clarify that to strengthen this section.
- Please clearly identify themes that you identified in this section.
- I would also consider adding which of the three aspects was identified the most and if the different aspects were more prevalent for students representing different demographics, like you did with the quantitative. You don’t need to include the numbers, but it would be beneficial to identify overall trends in qualitative data.
- Page 17, line 612
- Review grammar in this sentence
- Citations
- Starting on page 18
- Review citation formatting. Comma is missing in between the author/s and year.
- Page 18, line 662
- Year need the year
- Page 18, Line 684
- Make sure the citations are in ABC order
- Page 19, Lines 686-689
- Need a citation for this sentence.
- Discussion
- I would consider adding a figure to highlight your findings and implications of this work.
- The importance of your work can get lost in this section, so. Figure would allow you to emphasize what you found and how it relates to the field.
- Limitations and Future Research
- I would recommend adding this as sub-section in your discussion section.
- You need an explicit description of limitations to your current study, as well as recommendations for future research.
- Conclusion
- There is no clear conclusion to your article. I would also recommend adding a Conclusion section where you briefly, one paragraph explain what you learned in relation to the aims of this research. This will also allow you to emphasize why this work is important and how it adds to the field.
- Starting on page 18
- Participants and Setting
Author Response
Comment 1
Introduction- You discuss changes based on the pandemic, but many of the references you cite are pre-pandemic. I would recommend making sure to connect claims about shifts in how student wellbeing is viewed to post pandemic literature to strengthen that argument.
Thank you I have introduced the refs; Waite, P., McEachan, R. R. C., Pearcey, S., Jefferies, K., & Wright, J. (2024). Children’s behavioural and emotional wellbeing during the COVID-19 pandemic: Findings from the Born in Bradford COVID-19 mixed-methods longitudinal study. Journal of Child Psychology and Psychiatry, 65(5), 769–781. https://doi.org/10.1111/jcpp.13956 and; UNICEF Innocenti. (2025). Report Card 19: Child well-being in an unpredictable world. UNICEF Innocenti, Global Office of Research and Foresight. https://www.unicef.org/eca/press-releases/childrens-wellbeing-worlds-wealthiest-countries-took-sharp-turn-worse-wake-covid-19 in place of the refs (Henderson et al., 2020), Green et al., 2020,) line 42
2Comment 2: Page 2, line 41 Review formatting of citations
Thank you. (see new updated citations for post pandemic evidence; Waite et al., 2024 and UNICEF Innocenti, 2025 in place of previous early pandemic refs) We have checked and amended all citations
Comment 3: Page 2, line 57 Unclear why you used a footnote to cite the quote instead of a citation to stay consistent with journal formatting.
- Page 2, line 41
- Review formatting of citations
(see new updated citations for post pandemic evidence; Waite et al., 2024 and UNICEF Innocenti, 2025 in place of previous early pandemic refs)
- Page 2, line 57
- Unclear why you used a footnote to cite the quote instead of a citation to stay consistent with journal formatting.
- Same comment for quote on page 4, lines 142-143.
REsponses REmoved footnotes and added to references
- Please make sure you are adding a page number for direct quotes. I would check this throughout your manuscript.
-
- Same comment for quote on page 4, lines 142-143.
REmoved footnotes and added to references
Comment 4: Please make sure you are adding a page number for direct quotes. I would check this throughout your manuscript.
Checked and included throughout
Comment 5: Page 2, lines 68-69 What do you mean by “individualizing” and “moralizing”? I would expand on this statement more.
This sentence has been explained with a policy reference to support the claim
Comment 6:: Page 3, Line 86 Be mindful of spacing
Done
Comment 7: Page 3, line 107. Review grammar
Word order changed
Comment 8: Page 3, lines 115-116. You state “developed by” in your sentence and then add parentheses around the people you are mentioning. I would take their name out of parentheses and leave the year in parentheses.
Done
Comment 9: In-text citations. Please review journal policy for how to format citations with multiple authors to ensure you are consistent in the use of et al. and when to include the last names of all authors in your citation. With APA 7th edition you would use et al. for references with three or more authors.
Done
Comment 10: Page 3, Lines 116-117. What school interventions are you referring to in this sentence?
A couple of sentences have been added here to clarify the schooling interventions tests in the cited sources.
Comment 11: Page 3, lines 131-135. Please add citations to support this information.
Specific policies evidencing these claims have been inserted
Comment 12; Page 4, lines 144-151. I would add some examples of the principles/findings.
The key principles have been included, with a note indicating that the current paper’s aim is to report on the findings from this study.
Comment 13: Page 5, line 161-162. Please consider revising this definition for clarity. Making sure your readers understand how you define belonging is important. I am not sure what you mean by embracing the “social and well” in this sentence. The introduction before the quote needs greater clarity.
This was a wording error, we meant ‘as well’ not ‘and well’. The sentence has now been revised for clarity.
Comment 14: Page 4, line 176 Delete period before in-text citation
Done
Comment 15: Page 4 & 5, Lines 179-187. Consider separating this sentence into multiple sentences.
Done
Comment 16: Page 5, line 186: Be mindful of spacing within citation
Done
Comment 17: Page 5, line 216. Please review word choice
As table of covariates has been included this paragraph has now been removed to avoid repetition
Comment 18: Page 5, line 218. What does SEND stand for? Want to make sure this introduced for an international readership that my not use this acronym.
Please see page 6 Line 245 where we first use the term SEND. We have here included ‘special educational needs (SEND)’ so this acronym is explained for readers.
Comment 19: Participants and Setting
Please include sub-sections where you provide information about the participants and the setting in greater detail. You provide brief bits of this information in your overall Methodology section, but please provide this information in greater detail.
Knowing more about the participant demographics is important, especially because you make claims about students from “non-dominant groups” in your results. The reader needs to know more about the demographics of participants to support claims made in the results
We have now included information about Participants and Setting to provide a fuller account of the demographics and school contexts. This now includes detailed information about the 2,078 survey respondents and 76 focus group participants, along with the range of social characteristics collected (gender, sexuality, faith, SEND, neurodivergence, care-experience, migration status, socio-economic indicators). To support transparency, we have also added a Table which summarises all covariates collected and linked to the survey.
Comment 20 data analysis Please add a subsection explaining the data analysis used for both quantitative and qualitative data. This needs to be made much clearer to support replication and to inform the results.
We have now included a Data Analysis subsection which explicitly separates quantitative and qualitative procedures and specifies the study’s Sequential Explanatory Design (Ivankova, Creswell & Stick, 2006). This also includeds how bothe qantitative and qualitative data were analysed.
Comment 21: Page 6, Lines 239-240. In a research manuscript the heading should be Results. What you have would be a relevant sub-heading for the results section. Consider adding additional sub-headings within your Results section to highlight specific themes identified. You have a lot of information and the main points get lost in the this section.
Thank you. We have introduced the Subheading results and subheadings as suggested
comment 22: I would consider adding a table where you identify the identity aspects you are focusing on and how you define them. This would be a beneficial reference point for readers to refer to as you describe the results and discussion section. For example, this would be a nice accompaniment for Table 2 where you align factors with identity aspects.
Thank you, this is a very helpful suggestion, and we have now included a table , which has been positioned as you indicate, just before what was previously Table 2 (now table 3)
Comment 23
- Repeat same paragraph twice.
- Page 6, Lines 246-251
- Page 7, Lines 254-259
Thank you, this second paragraph has now been removed.
Comment 24: Consider adding sub-headings based on the three aspects that were identified in the qualitative data. It is unclear from the current methods section how this data was analyzed, so you want to clarify that to strengthen this section.
We have now included a Data Analysis subsection which explicitly separates quantitative and qualitative procedures and specifies the study’s Sequential Explanatory Design (Ivankova, Creswell & Stick, 2006). This also includeds how bothe qantitative and qualitative data were analysed.
Comment 25: Please clearly identify themes that you identified in this section. I would also consider adding which of the three aspects was identified the most and if the different aspects were more prevalent for students representing different demographics, like you did with the quantitative. You don’t need to include the numbers, but it would be beneficial to identify overall trends in qualitative data.
Sub-headings have now been added to this section (now labelled 3.1.2 as there had been a repetition of the previous subtitle). Themes have been added and the prevalence of the themes for different groups of students has been signposted.
Comment 26: Page 17, line 612 Review grammar in this sentence
Amended
Comment 27: Review citation formatting. Comma is missing in between the author/s and year
done throughout
Comment 28: Page 18, line 662. Need the year
added
Comment 29 Make sure the citations are in ABC order
done
Comment 30: Page 19, Lines 686-689 Need a citation for this sentence.
Done.
Comment 31 Discussion I would consider adding a figure to highlight your findings and implications of this work. The importance of your work can get lost in this section, so. Figure would allow you to emphasize what you found and how it relates to the field
Thank you for this very helpful suggestion. A figure has now been added to the discussion
Comment 32: Limitations and Future Research I would recommend adding this as sub-section in your discussion section. You need an explicit description of limitations to your current study, as well as recommendations for future research.
Good suggestion. This section has now been added.
Comment 33 Conclusion There is no clear conclusion to your article. I would also recommend adding a Conclusion section where you briefly, one paragraph explain what you learned in relation to the aims of this research. This will also allow you to emphasize why this work is important and how it adds to the field.
Thank you this has now been included.
Round 2
Reviewer 2 Report
Comments and Suggestions for Authors
Thank you for the opportunity to review your revised manuscript. I appreciated the detailed responses you provided to explain how you addressed reviewer feedback. The additional information strengthened the methods section and provided greater clarity about the participants and setting, as well as data analaysis. The discussion has also been strengthed with the addition of a limitations and future research section. The conclusion and figure provide a clear explanation fo the relevance of this work and how it contributes to the field.
Recommendations for authors:
- You added a lot of beneficial information to the Methods section. I would encourage you to add additional headings to separate the important components of this section, such as as Participants, Setting, Procedures, and Measures. This will provide greater clarity about these essential components and support replication.
- You strengthed the reader's understanding of your Data Analysis by adding this sub-section. For your qualitative analysis paragraph, please add literature that informed your data analysis process. This will strengthen this methodology.
Author Response
- You added a lot of beneficial information to the Methods section. I would encourage you to add additional headings to separate the important components of this section, such as as Participants, Setting, Procedures, and Measures. This will provide greater clarity about these essential components and support replication.
Response 1: Thank you. We have included the suggested subheadings across pages 6, 7 and 8
- You strengthened the reader's understanding of your Data Analysis by adding this sub-section. For your qualitative analysis paragraph, please add literature that informed your data analysis process. This will strengthen this methodology.
Response 2: Thank you. We have inserted a citation to two relevant papers and added these to the references.